# The Jacalin-Related Lectin HvHorcH Is Involved in the Physiological Response of Barley Roots to Salt Stress

**DOI:** 10.3390/ijms221910248

**Published:** 2021-09-23

**Authors:** Katja Witzel, Andrea Matros, Uwe Bertsch, Tariq Aftab, Twan Rutten, Eswarayya Ramireddy, Michael Melzer, Gotthard Kunze, Hans-Peter Mock

**Affiliations:** 1Leibniz Institute of Vegetable and Ornamental Crops, 14979 Großbeeren, Germany; 2Leibniz Institute of Plant Genetics and Crop Plant Research, Stadt Seeland, 06466 Gatersleben, Germany; andrea.matros@julius-kuehn.de (A.M.); uwe.bertsch@louisenlund.de (U.B.); tarik.alig@gmail.com (T.A.); rutten@ipk-gatersleben.de (T.R.); melzer@ipk-gatersleben.de (M.M.); kunzeg@ipk-gatersleben.de (G.K.); mock@ipk-gatersleben.de (H.-P.M.); 3Dahlem Centre of Plant Sciences, Institute of Biology/Applied Genetics, Free University of Berlin, 14195 Berlin, Germany; eswar.ramireddy@iisertirupati.ac.in

**Keywords:** abiotic stress, apoplast, functional screening, *Hordeum vulgare*, salinity, seedling

## Abstract

Salt stress tolerance of crop plants is a trait with increasing value for future food production. In an attempt to identify proteins that participate in the salt stress response of barley, we have used a cDNA library from salt-stressed seedling roots of the relatively salt-stress-tolerant cv. Morex for the transfection of a salt-stress-sensitive yeast strain (*Saccharomyces cerevisiae* YSH818 *Δhog1* mutant). From the retrieved cDNA sequences conferring salt tolerance to the yeast mutant, eleven contained the coding sequence of a jacalin-related lectin (JRL) that shows homology to the previously identified JRL horcolin from barley coleoptiles that we therefore named the gene *HvHorcH*. The detection of HvHorcH protein in root extracellular fluid suggests a secretion under stress conditions. Furthermore, HvHorcH exhibited specificity towards mannose. Protein abundance of HvHorcH in roots of salt-sensitive or salt-tolerant barley cultivars were not trait-specific to salinity treatment, but protein levels increased in response to the treatment, particularly in the root tip. Expression of HvHorcH in *Arabidopsis thaliana* root tips increased salt tolerance. Hence, we conclude that this protein is involved in the adaptation of plants to salinity.

## 1. Introduction

The area of farmland worldwide that can be cultivated with crop plants is declining because of increasing drought periods and the increasing salinity of the soil in several regions of the earth [1]. Global climate change, which is predicted to be accompanied by prolonged and intensified drought periods, is likely to aggravate this situation even further. Intensified irrigation attempts to combat drought ultimately increase soil salinity and thus eventually impede farmland cultivation when salinity reaches threshold levels that can no longer be tolerated by crop plants [2]. It is therefore an eminent goal for a global sustainable food supply to improve the salt stress tolerance of crop plants in order to push these thresholds of soil salinity upwards so that more farmland with high-salinity soil will still be amenable to agriculture.

Barley is regarded as one of the more salt-stress-tolerant crop species [3,4], although the level of tolerance may vary considerably between diverse barley cultivars [5]. Therefore, the comparison of proteomes and transcriptomes of contrasting barley cultivars with regard to their salt stress tolerance bears the potential to identify candidate proteins and their respective coding and regulatory genes that play a role in salt stress tolerance of barley. A collection of salt stress candidate genes discovered by a proteomics approach using different barley cultivars has been published previously [6,7,8,9]. However, the profiling of low-abundant, low-solubilizing, or small proteins in untargeted approaches is still challenging due to intrinsic technical limitations of proteomic analyses. Hence, we constructed a cDNA library made from the salt-stressed roots of the more tolerant barley cv. Morex to search for barley proteins related to stress tolerance using a functional screening procedure that involved a salt-stress-susceptible yeast mutant. The basis for the yeast complementation screening approach applied here is formed by a global commonality of the signal transduction pathways and mechanisms responsible for osmotic stress tolerance in both yeasts and plants [10,11,12,13,14]. A yeast screen of a cDNA library generated from salt-stressed sugar beet leaves led to the identification of a serine O-acetyltransferase conferring yeast osmotolerance [15]. In addition, new candidates for salt stress response were obtained by screening a date palm [16] and a *Thellungiella halophila* cDNA library [17]. The feasibility of this approach for *Escherichia coli* was also shown. The expression of a cDNA library derived from salt-treated *Solanum commersonii* in *E. coli* identified novel genes involved in the stress response [18,19]. A lectin receptor-like kinase was isolated from screening an *E. coli* expression library generated from salt-stressed pea seedlings [20].

Using a similar salt stress tolerance complementation assay, we identified cDNAs for a jacalin-related lectin (JRL) with homology to the JRL horcolin from barley coleoptiles. The corresponding protein was isolated from barley roots based on its mannose affinity and characterized regarding its salt-stress-induced expression in barley root tissues as well as in *Arabidopsis thaliana*. HvHorcH1 association with barley salt stress tolerance was carried out by studying the HvHorcH1 protein expression levels in salt-sensitive and salt-tolerant barley cultivars. Further, transgenic Arabidopsis with root-tip specific expression of HvHorcH1 were generated.

## 2. Results

### 2.1. Screening for Salt Stress-Related Genes from Barley cv. Morex Roots in Salt Sensitive Yeast

In order to identify gene products that participate in the salt stress tolerance of barley, we used a cDNA library from salt-stressed roots of the relatively salt-stress-tolerant cv. Morex for the transformation of the salt-stress-sensitive yeast strain YSH818. A total of 51 independent yeast transformants, containing 21 different barley cDNAs, were isolated that showed increased salt tolerance as compared to the empty vector control (Table 1). Several cDNAs represented complete open reading frames, but also cDNAs lacking 5′ or 3′ ends were found. For further validation, cDNAs were used for the retransformation of YSH818, and salt stress testing was extended up to 3% NaCl in the growth medium. Figure 1 shows exemplarily the results for two jacalin-related lectin 31 (JRL) cDNAs (09_818-20-10-10-5, 37_818-28-8-10-5, both coding for HORVU7Hr1G059330), where a vector control is contrasted to the transformed cells. Here, the JRL31-containing yeast cells show clearly increased survival at higher salt concentrations of 2% and 3% NaCl, even at higher dilutions of the initial inoculum, when compared to the vector control.

Currently, the putative function of several of the isolated cDNAs is unknown (Table 1). Other functional groups found in the analysis included cDNAs involved in translation, cellulose biosynthesis, DNA binding, fatty acid metabolism, vesicle transport, proteolysis, and stress response, among others. Ten barley cDNA sequences were retrieved that all were highly homologous to the coding sequence of a yet unannotated barley JRL31 HORVU7Hr1G059330, which shows a 50% sequence identity with the previously identified mannose-specific JRL horcolin (HORVU1Hr1G000160) from barley coleoptiles [21].

Appendix A shows an amino acid sequence alignment of those sequences. Hence, the isolated JRL31 were termed as horcolin homolog HvHorcH (HORVU7Hr1G059330).

A further interesting candidate identified in the screen, besides JRL31, was cellulose-synthase like D2 (HORVU2Hr1G042350), identified in six independent yeast transformants. However, for all further experiments, we focused on JRL31 cDNAs.

### 2.2. In Silico Gene Expression Pattern of HvHorcH in Barley Roots

The expression level of *HvHorcH* gene transcripts was analysed using the Expression Atlas (Expression Atlas. Available online: https://www.ebi.ac.uk/gxa/home, accessed on 24 August 2021), a repository of RNAseq expression data, which is reproduced in Appendix A, and compared to that of horcolin. The highest transcript levels for horcolin were found for shoots and germinated embryos. The distribution pattern of HvHorcH was characterized by an accumulation mainly in roots, indicating a different functionality as compared to horcolin.

### 2.3. Cellular and Subcellular Localization of HvHorcH

Based on transcript information, HvHorcH is expressed mainly in roots. In order to investigate the localization within the root, we used the polyclonal HvHorcH antibody, in combination with a fluorescently labeled anti-rabbit-IgG antibody, in the salt-stressed roots of cv. Morex. A non-uniform tissue distribution was observed with a strong labeling of the root cap (Figure 2).

Localization in the apoplast has been described for the homologous horcolin [21] as well as for the JRL helja from *Helianthus annuus* seeds [22]. In order to study the subcellular distribution of HvHorcH, we prepared an apoplast fraction of salt-stressed roots of cv. Morex. The presence of HvHorcH in apoplastic fluid of salt-stressed roots of cv. Morex was confirmed immunologically with antibodies specific for HvHorcH. The apoplastic fluid contained only minor contamination with cytosolic proteins, as exemplified by the presence of glyceraldehyde-3-phosphate dehydrogenase (GAPDH, Figure 3). Since HvHorcH lacks a signal sequence for an export via the standard secretory pathway through ER and Golgi, its secretion into the apoplast of roots may follow a non-canonical route.

### 2.4. Mannose-Binding of HvHorcH Protein

In order to test whether HvHorcH has a similar binding behaviour to mannose as horcolin, a mannose affinity chromatography was performed. HvHorcH was enriched from salt-stressed cv. Morex roots to apparent homogeneity in a two-step purification process employing anion exchange (AEX) and mannose affinity chromatography. Fractions from this purification procedure were evaluated by immunoblotting using an anti-HvHorcH polyclonal antiserum and by SDS-PAGE (Appendix A). A protein band with a molecular weight (MW) of about 15 kDa, which corresponds to the theoretical MW of HvHorcH of 15.1 kDa, was detected both in the native root extract (R) and after desalting and buffer exchange (R_d_). Subsequent AEX chromatography resulted in the enrichment of HvHorcH in a number of fractions during elution with a linear NaCl gradient (Appendix A, F_13_ to F_16_). These fractions were combined (P) and subjected to dialysis (P_d_). By the following mannose affinity chromatography, a fraction highly enriched in HvHorcH could be generated. Thereby, the majority of the target protein eluted within the first fraction (E1_1_) of the first elution with citrate buffer (elution buffer 1). For the recombinant expressed HvHorcH protein, a signal in the range of 15 to 18 kDa has been detected, which can be explained by the attached poly-histidine affinity tag.

### 2.5. Selection of Barley Lines with Different Salt Stress Tolerance Levels

In order to test whether HvHorcH expression levels are associated with barley salt stress tolerance, we screened doubled-haploid (DH) offspring lines of the cross between the more tolerant cv. Morex and the more sensitive cv. Steptoe. The latter has been previously screened for salt tolerance in a germination assay [23]. DH lines showing a more sensitive response such as cv. Steptoe or a more tolerant one as compared to cv. Morex were evaluated in a hydroponic assay for salt tolerance at the seedling stage. Previously, we found that the development of the third leaf was indicative of a contrasting response toward salinity [8]. Hence, this parameter was used to assess the response of tested accessions. Four accessions were isolated showing a comparable contrasting response to the parental lines cvs. Steptoe and Morex with respect to the length and fresh weight of the third leaf (Figure 4a,b). With increasing NaCl concentration, the growth was inhibited in all accessions, but the growth inhibition of DH14 and DH43 was stronger at all levels tested as compared to DH187 and DH198. The leaf fresh weight at 150 mM NaCl was reduced to 10% and 8% in DH14 and DH43, respectively, and to 27% and 46% in DH187 and DH198, respectively, as compared to the control treatment. Under the same conditions, the remaining growth rate was 4% in cv. Steptoe and 41% in cv. Morex [8], indicating that the response of selected offspring lines was comparable to that of the parent cultivars. In our previous work, we observed pronounced differences of salt-stress-mediated alterations in root architecture between the tolerant cv. Morex and the sensitive cv. Steptoe [6]. Additionally, the transcript levels of *HvHorcH* (Appendix A) and the protein levels (Figure 2) were shown to be highest in roots.

Thus, we additionally investigated root architecture responses in the selected DH lines under control conditions and salt stress (Figure 4c). Morphological parameters (branches, ends, and crossings) were evaluated after ten days exposure to 150 mM NaCl from digital root images and displayed as a function of the distance from the root base (Figure 4d). Under control conditions, the more tolerant cv. Morex, DH187 and DH198, displayed a more elaborate root system containing higher numbers of ends and crossings in greater distance from the root base (>100 mm, control). Additionally, the roots of those lines were more branched far from the root base under the control treatment. When grown under salt stress, all lines tested showed a strong shortening of roots and a high reduction in number of branches, ends, and crossings. Nevertheless, the highest capability for maintaining root growth and differentiation was observed for the salt-tolerant accession DH198, indicated by the longest and most branched root system under salt stress (Figure 4d).

### 2.6. HvHorcH Protein Abundance Is Highly Increased in Root Tips under Salt Stress

The distribution of HvHorcH across the roots of the six selected barley lines was analyzed using the polyclonal antibody against HvHorcH. Protein abundance in roots and root tips after salt stress imposition as compared to control conditions was investigated by protein gel blot analysis (Figure 5). The antibody clearly detected a protein in the molecular weight range of 15 kDa, matching the calculated molecular weight of 15.1 kDa for HvHorcH. Under control conditions, all accessions (except DH14) accumulated higher amounts of HvHorcH in total roots than in root tips. A marked increase in HvHorcH abundance was shown for all accessions when grown at 150 mM NaCl. Notably, this increase was much higher in root tips as compared to total roots, indicated by higher fold change values for all accessions in this tissue. However, no concerted up-regulation of the protein in the salt-tolerant or salt-sensitive accessions was observed for roots and root tips. Thus, we assume that HvHorcH1 is involved in the general osmotic response mechanisms during growth under salt stress in barley.

### 2.7. HvHorcH Enhances Salt Tolerance in Transgenic Arabidopsis

To further study the function of *HvHorcH* during salt stress, a transgenic approach in *Arabidopsis thaliana* was applied. Since the protein showed a root-tip-specific expression under salinity in barley, a root-tip-specific promoter was selected to drive *HvHorcH* gene expression, coupled to eGFP as the reporter gene. Three independent lines expressing *HvHorcH* in the Col-0 background were selected for further studies based on the strength of the reporter signal (Appendix A). When grown in the presence of different levels of NaCl, the fresh mass production of the rosette in the transgenic lines 3-8, 5-7 and 9-6 was more advanced than in the wildtype Col-0 (Figure 6a). When root phenotypes of plants grown in the presence of 50 mM NaCl were analyzed, an increased primary root length of transgenic lines was observed as compared to wildtype plants (Figure 6b). These results indicate that expression of HvHorcH in root tips enhances salt tolerance in *A. thaliana*.

## 3. Discussion

The functional screening of barley salt-stress-related genes in yeast showed that, to a certain extent, plant proteins can substitute for components of the stress signaling pathway or the cellular defense mechanisms that are blocked in the *Δhog1*-mutant yeast used in this study. One of the cDNAs conferring salt tolerance to yeast encoded the membrane steroid binding protein 1 (MSBP). In our group, this protein was recently identified in a comparative root plasma membrane proteomic analysis of barley cvs. Steptoe and Morex to be involved in the adaptation of root architecture to salinity [6]. However, numerous genes isolated in our present study are not yet described to be involved in the barley salt stress response and hence represent novel candidates for further evaluation.

Most multiple cDNA isolations were derived for a JRL 31 (named HvHorcH), which was therefore selected for further analysis. HvHorcH belongs to the group of jacalin-related lectins (JRLs), which are carbohydrate-binding proteins. In general, lectins cover a diverse group of protein families with multiple functions that bind reversibly to specific mono- or oligosaccharides without altering the substrate [24]. JRLs are a subgroup that share one or more conserved domains with jacalin, a lectin isolated from jackfruit [25]. Some JRLs are involved in the response towards biotic stresses [26,27,28], while others play a role in developmental processes [29]. Numerous studies have noticed the response of JRLs to abiotic stresses, such as high light [30] and salt stress [31,32]. In barley, several JRLs are described with different domain organization and expression patterns. Lem2 is a JRL with two jacalin domains, which is expressed in shoots, and its induction by salicylic acid suggests an involvement in systemic acquired resistance [33]. Chimeric JRLs JRG1.1, JRG1.2, and JRG1.3 (dirigent protein) consist of one jacalin and one dirigent domain. They are assumed to be involved in plant disease response since inducibility by jasmonic acid was demonstrated [34]. Chitin-binding ability was shown for BLc3, a lectin that accumulates in root tips and in the embryo [35].

The proteins HL2 and horcolin are characterized by a single jacalin domain. HL2 is expressed in shoots and transcripts accumulate under high light conditions and after methyl jasmonate treatment [30]. In contrast to this, horcolin transcript and protein levels are induced specifically in etiolated coleoptiles, and the protein is localized to the cell wall and can be extracted in apoplastic washing fluid [21]. HL2 and horcolin belong to the subgroup of mannose-binding JRL, a property also demonstrated for HvHorcH in our study. We have detected HvHorcH in apoplastic fluid preparations from roots, indicating a possible rhizodeposition of this protein, a hypothesis that is further underlined by the root-tip-specific accumulation of HvHorcH under salt stress conditions. This accumulation pattern was observed for all investigated genotypes, pointing toward a role of HvHorcH in the general osmotic response mechanisms during growth under salt stress in barley.

In addition, the mannose-specific JRL helja from *H. annuus* was localized to the seedling apoplast, where its release protects the roots from pathogenic fungi through cell agglutination [22,36,37]. A role in plant defense against pathogens was also proposed for chitin-binding JRL found in barley root tips, which share a high sequence similarity to wheat germ agglutinin [35,38,39]. In contrast to this, reports on wheat germ agglutinin itself underline its role in establishing ABA-controlled unspecific defense responses since they accumulate in wheat roots in response to drought stress [40] and salt stress [41], while exogenous wheat germ agglutinin is able to protect the seedling from salt-stress-induced growth arrest [41]. While the expression of JRL in the meristem of drought-stressed banana is reduced [42], the levels of rice JRL salT increase in response to drought and salt stress, although mainly in the shoot [43,44]. When testing rice cultivars of contrasting salt tolerance, salT protein accumulated nonspecifically, indicating no correlation between salt stress response and protein levels [45]. Further, the rice JRL MRL was also inducible after imposition of salt stress and even higher in roots as compared to shoots [46].

As we have found in our study that the accumulation of HvHorcH in response to salt stress is not correlated to the tolerance or sensitivity of the tested barley genotypes, we also assume that HvHorcH has a general protective role in the root tip. A non-uniform distribution was observed with strong labeling of the root cap, a similar distribution to that found for other root-specific lectins [35,38]. The rhizodeposition of HvHorcH by root tips could modify the osmotic potential in its vicinity in order to maintain meristematic activity under stress conditions, as shown for wheat germ agglutinin in wheat [47] and barley [48]. Recently, two JRL were detected in extracellular vesicles isolated from the *A. thaliana* leaf apoplast, indicating that JRL are secreted by extracellular vesicles [49]. We have shown here that HvHorcH is functional in *A. thaliana* root tips and that transgenic plants exhibit a greater tolerance towards salt stress. However, the precise functional mechanism behind the protective role of HvHorcH in roots remains to be elucidated. To examine the binding specificities of HvHorcH more profoundly, carbohydrate microarrays should be performed next [50].

A further candidate identified in our yeast screen was a cellulose-synthase-like D2 gene. This gene is 99% identical to cellulose-synthase-like *HvCslF3*, a gene involved in the biosynthesis of (1,3;1,4)-*β*-D-glucan, which is the major cell wall constituent in monocots and is the most abundant structural polysaccharide of the mature barley grain [51]. So far, ten *HvCslF* genes are reported [52], with *HvCslF3* not being involved in grain development, based on transcript profiling data [51,53] but being highly expressed in the coleoptile and root tips [52]. One of the most devastating effects of salt stress in plants is the induction of detrimental changes in the cell wall, e.g., disturbed cellulose synthesis [54], and reduced pectin cross-linking and cell wall integrity [55]. Often, an altered composition of cell wall components in response to salt stress is observed [56], although the precise role of (1,3;1,4)-*β*-D-glucan in stress adaptation remains to be elucidated. Notably, the function of C-type lectins as receptors of linear (1,3)-*β*-D-glucan and its role in triggering diverse immune responses in humans, invertebrates, and microorganisms has been established [57]. Similarly, the chitin elicitor receptor kinase 1 (CERK1), initially reported to directly bind chitin oligosaccharides, has been suggested to function as a co-receptor for linear (1,3)-*β*-D-glucan, triggering fungal immune responses in *A. thaliana*, recently [58]. Recently, the multivalent binding of horcolin to high-mannose glycans has been demonstrated [59]. Whether a similar function may be assigned to HvHorcH remains speculation at this point. However, the observed longer primary roots in *HvHorcH* expressing *A. thaliana* and the larger root systems of salt-tolerant barley genotypes hint toward effective salt stress tolerance mechanisms related to maintaining root growth under saline conditions in both species, which may involve HvHorcH and HvCslF3. In our study, we have used two groups of barley genotypes with contrasting salt stress response to distinguish specific or nonspecific stress responses. The same genotypes should now be tested for in-depth characterization of HvCslF3 and other candidates isolated by our yeast screen, such as those involved in stress response or ion transport in order to identify stress-response-related genes for plant improvement.

## 4. Materials and Methods

### 4.1. Plant Material and Salt Stress Treatment

Barley cultivars Steptoe, Morex as well as their offsprings DH14, DH43, DH187, and DH198 were grown in hydroponic culture as described by Witzel et al. [8]. Salt stress treatment began six days after germination by the NaCl concentration being increased step-wise (50 mM NaCl steps every other day) until 150 mM NaCl. Root material was harvested at ten days for cDNA library construction, at five and ten days for qPCR measurements, and at seven days for immunodetection, after this final level had been reached. Material from five individual plants was combined to form, for each of the cultivars, one batch of non-treated (control) roots, and a second one of salt-stressed roots. Three independent experiments were performed. For the screening of doubled-haploid lines of the Steptoe × Morex population, plants were grown as described above but supplied with 50, 100, 150, 200, or 250 mM NaCl for ten days. Seventeen to twenty plants per genotype and treatment were sampled, and shoot biomass was recorded. This experiment was performed once.

To assess the transgene effect on plant biomass production, *A. thaliana* wild-type Col-0 and the same three transgenic lines were grown on non-sterile standardized plant growth substrate (Fruhstorfer Erde type P, Hawita, Vechta, Germany) with a pH of 6.0 in a climate chamber under short-day conditions (8 h light/16 h dark, 22 °C, 40–60% humidity, 300 μmol m^–2^ s^–1^). After 2 weeks, plants were transferred into single pots and cultivated for further two weeks. Salt stress treatment was performed as described above for barley, with increasing concentration in a step-wise manner up to 150 mM NaCl and maintained for one week. At harvest, the rosette fresh weight of 25–27 plants per treatment and genotype was measured. These experiments were performed in triplicate. Statistical evaluation of rosette biomass was calculated using paired t-test, implemented in Sigma Plot 14.0 (Systat Software, Frankfurt am Main, Germany).

For analyzing root morphological effects, seeds of *A. thaliana* wild-type Col-0 and three transgenic lines expressing *HvHorcH* were surface sterilized. Seeds were germinated and grown on half-strength Murashige–Skoog (MS) medium (pH 5.8) and supplemented with 1.5% *w/v* sucrose for 2 weeks. Then, seedlings were transferred to a medium supplemented with or without 50 mM NaCl. Plants were maintained at 18/20 °C under an 8 h light/16 h dark rhythm (300 μmol m^–2^ s^–1^) for one week. These experiments were performed in triplicate.

### 4.2. Yeast Strain and Cultivation

The salt-sensitive yeast *Saccharomyces cerevisiae* strain YSH818 (MATa *leu2-3/112 ura3-1 trp1-1 his3-11/15 ade2-1 can1-100 GAL SUC2 hog1Δ::LEU2*) [60] was used for barley cDNA testing. Yeast cultures were grown under selective conditions on synthetic dextrose minimal medium (0.67% bacto-yeast nitrogen base without amino acids, 1.6% bacto-agar) at 30 °C with 2% glucose as carbon source and the addition of amino acids [9].

### 4.3. CDNA Library Construction and cDNA Isolation

A cDNA expression library was constructed in the *S. cerevisiae* strain YSH818 using the Clone Miner cDNA Library Construction Kit (Thermo Fisher Scientific, Waltham, United States), according to the manufacturer’s instructions. RNA was extracted from roots of salt-stress-treated barley cv. Morex using the RNeasy Plant Mini Kit (Qiagen, Hilden, Germany), and mRNA was enriched by using the polyA Spin™ mRNA Isolation Kit (New England Biolabs, Ipswich, MA, United States). A total of 2 µg of mRNA was used for cDNA library construction, leading to the integration of cDNAs into pDONOR222 vector. The expression library (8 × 10^5^ colony forming units, average insert size of 950 bp) was subsequently shuttled into pYES-DEST52 vector for yeast transformation, as described before [9]. Transformed yeast cells were plated onto the medium as described above, with additional 2% NaCl. Transformants with increased salt tolerance after four days of growth, as compared to the empty vector control, were isolated; cDNA was extracted, sequenced, and used for retransforming YSH818 control cells. For the re-evaluation of cDNAs, 2% and 3% NaCl were added to the medium to impose salt stress.

### 4.4. Database Search and Sequence Alignments

Barley cDNAs that were extracted from yeast cells were sequenced, and sequences were compared to barlex CDS_HC May2016 database [61] using blastx (BARLEX. https://apex.ipk-gatersleben.de/apex/f?p=284:10, accessed on 24 August 2021). The amino acid sequence alignment was performed using the T-Coffee program and clustalW algorithm [62]. A conserved domain database search was performed using NCBI’s protein–protein blast [63].

### 4.5. Root Morphological Analysis

After a ten-day exposure to either control conditions or 150 mM NaCl, the plant roots of all six barley cultivars were transferred to a flat water-filled tray. After detangling the roots using a toothpick, images were captured by a conventional digital camera. Root structure was determined from the images. Branch points (branches), end points (ends), and points of intersection (crossings) were detected as described [6]. Numbers of branches, ends, and crossings were graphically displayed as a function of the distance from the root base in mm by a Sholl analysis [64]. Three biological experiments were performed, and for each cultivar and treatment, the roots of four to five plants were analyzed, respectively.

### 4.6. Expression of Recombinant HvHorcH in Escherichia coli

For expression in *E. coli*, the barley EST clone GCW003B01r, coding for HvHorcH, was retrieved from CR-EST database (CR-EST. https://apex.ipk-gatersleben.de/apex/f?p=CREST:1:14690193541182::NO:::, accessed on 11 June 2009) and introduced into the pCR 2.1. Topo vector (Thermo Fisher Scientific, USA). By PCR, a SalI site at the 3′-end and a BamHI site at the 5′-end were introduced. The resulting fragment was cloned into the pQE30 expression vector by using the QIAexpress Type IV Kit (Qiagen, Hilden, Germany) and transformed into *E. coli* strain XL1Blue (Stratagene, Amsterdam, The Netherlands). Recombinant HvHorcH was purified using affinity chromatography on Ni-NTA agarose (Qiagen, Hilden, Germany) following the manufacturer’s instructions. For immunodetection, polyclonal antibodies were raised against recombinant HvHorcH in rabbits.

### 4.7. Protein Extraction, SDS-Polyacrylamide Gel Electrophoresis (PAGE), and Gel Blot Analysis

Proteins were isolated from total roots and root tips from plant material harvested seven days after final salt stress (150 mM NaCl) application. The root material from five plants of each genotype was combined for protein extraction, and all the experiments were performed in triplicates. The frozen root material was homogenized under liquid nitrogen to a fine powder, mixed with TCA/acetone solution (10% (*w*/*v*) TCA, 0.07% (*w*/*v*) 2-mercaptoethanol in acetone) in a ratio of 100 mg to 1 mL, and incubated for 45 min at –20 C°. The precipitate was pelleted by centrifugation, washed twice with 0.07% (*w*/*v*) 2-mercaptoethanol in acetone, and dried in a vacuum centrifuge. Protein precipitates were dissolved in loading buffer (56 mM Na_2_CO_3_, 56 mM dithioerythritol, 0,1% (*w*/*v*) sodium dodecyl sulfate, 12% saccharose, and 0,01% bromophenol blue). Quantification was performed using a Bradford assay (Bio-Rad, Hercules, CA, United States), using bovine serum albumin as a standard.

Samples of each extract with equal amounts of protein (2 µg) were subjected to SDS-polyacrylamide gel electrophoresis [65] on 15% acrylamide gels and transferred to an Immobilon-P PVDF membrane (Merck Millipore, Darmstadt, Germany) using a semi-dry blotting apparatus (Schütt, Göttingen, Germany). Membranes were probed with anti-HvHorcH polyclonal antiserum followed by the incubation with infrared dye-coupled secondary antibody (Li-Cor, Bad Homburg, Germany). Documentation and quantification was performed on a Li-Cor scanner driven by Odyssey software v3.0 (Li-Cor, Bad Homburg, Germany). The MagicMark™ XP Western Protein Standard (Thermo Fisher Scientific, Waltham, MA, United States) served as the protein standard.

### 4.8. Microscopy

For immune-histochemical examination, 2 mm long root tip sections were collected. After microwave-assisted fixation and LR White resin infiltration using a PELCO e BioWave^®^ Pro^+^ (TedPella, Redding, CA, United States), according to Appendix A, resin was polymerized at 60 °C for 48h.

Median longitudinal root sections of 0.5 µm thickness were cut on an Ultracut UCT instrument (Leica, Nussloch, Germany) and mounted on 8-well poly-L lysine coated slides (Thermo Scientific, Waltham, CA, United States). To avoid non-specific antibody binding, sections were blocked at room temperature (RT) for 20 min in PBS buffer containing 3% (*w/v*) bovine serum albumin and 0.1% Tween, followed by washing with washing buffer (0.1% bovine serum albumin and 0.05% Tween 20 in PBS) for 2 min at RT. After primary antibody incubation with polyclonal rabbit anti-HvHorcH (see Section 4.7, 1:10 and 1:100 diluted) for 45 min at 37 °C, probes were washed five times for 5 min at RT with PBS buffer. Incubation with secondary antibody goat anti-rabbit Alexa fluor 488 (MoBiTec, Göttingen, Germany; dilution 1:200) for 30 min at 37 °C was followed by five final washes with a washing buffer for 5 min at RT. Sections were mounted in antifade (1 mL 0.1 M Tris, pH 9.0, 9 mL glycerin, 50 mg n-propyl gallate). Immunolabeling was analyzed using a LSM 510META confocal microscope (Carl Zeiss, Jena, Germany). Alexa 488 was visualized with a 488 nm laser line in combination with a 505–530 nm bandpass filter. For controls, sections were incubated with Alexa 488 only.

### 4.9. Collection of Apoplastic Fluid

Apoplast infiltration with 1 × phosphate buffered saline of pH 7.0 containing 300 mM mannose was carried out by submerging the roots in the buffer in a 50 mL falcon tube and infiltrating under a vacuum of exactly 20 mPa generated by a motor pump for 30 s. The centrifugation of the surface-dried roots was carried out in a 5 mL syringe placed within a 15 mL falcon tube at a centrifugation speed of 400 × *g*. The presence of HvHorcH and GAPDH (Agrisera AB, Vännäs, Sweden) in the respective fractions was validated by immunoblotting, as described before.

### 4.10. Affinity Enrichment of HvHorcH by Mannose Binding

Roots from salt-stressed cv. Morex plants were ground to a fine powder under liquid nitrogen. Proteins were isolated on ice by grinding the root material with a cold extraction buffer (20 mM tris(hydroxymethyl) aminomethane (Tris) HCl, pH 7; 300 mM mannose; 1 mM dithiothreitol, and with protease (cOmplete™) and phosphatase inhibitor (PhosSTOP™, Roche Diagnostics GmbH, Mannheim, Germany)) in a ratio of 2.5 g to 4 mL. Insoluble material was sedimented by centrifugation (10,000× *g*, 4 °C, 15 min). The supernatant was transferred to a fresh tube and the pellet re-extracted with 2 mL of cold extraction buffer. After centrifugation (10,000× *g*, 4 °C, 15 min), both supernatants were combined, resulting in about 5 mL of crude extract. The extract was desalted, small contaminants removed, and the buffer exchanged to an anion exchange buffer (20 mM 2-(N-morpholino)ethanesulfonic acid, 1 mM dithiothreitol, adjusted to pH 6.5 with 1 N NaOH) by gel filtration on PD10 columns (GE Healthcare, Chicago, IL, USA) as instructed by the manufacturer. After elution from PD10 columns, an extract volume of about 7 mL was reached. Anion exchange chromatography was performed on a 6 mL Ressource™Q column (GE Healthcare) equilibrated with five column volumes (CV) of anion exchange buffer and utilizing an automated Äkta start protein purification system (GE Healthcare, Chicago, IL, USA). The protein extract was loaded on the column in anion exchange buffer at a flow rate of 2 mL/min. Sequential elution of bound proteins was realized by a linear gradient from 0 to 500 mM of NaCl in anion exchange buffer, and fractions of 3 mL each were collected. SDS-PAGE and protein gel blot analysis of the purified fractions were performed as stated above. Fractions containing HvHorcH were combined (about 18 mL) and subjected to dialysis (4 °C, overnight) against a minimum of 40 volumes of affinity chromatography buffer (50 mM Tris HCl, pH 7.2; 100 mM NaCl; 1 mM MgCl_2_) utilizing 2 mL PlusOneTM Mini Dialysis Kit tubes with a 1 kDa cut-off (GE Healthcare, Chicago, PD, United States) according to the manufacturer’s instruction. The resulting fraction volume was stable with about 18 mL. Affinity chromatography was performed using D-mannose agarose (Sigma-Aldrich, United States) with a protein-binding capacity of about 40 mg/mL. A 20 mL empty Poly-prep^®^ gravity-flow column (Bio-Rad, Hercules, CA, United States) was filled with 0.6 mL of D-mannose agarose matrix and equilibrated with 20 CV of affinity chromatography buffer. The dialyzed protein extract (18 mL) was added and the column sealed and incubated at 4 °C on an overhead shaker for a minimum of one hour. The column was then placed on a gravity stand and the flow-through was collected. Unbound proteins were washed off with 13 CV of affinity chromatography buffer (8 mL). Then, two times 1 mL of elution buffer 1 (50 mM MES, pH 6.5 adjusted with 1 N NaOH, 300 mM D-mannose, and with protease (cOmplete™, Roche Diagnostics GmbH, Mannheim, Germany) and phosphatase inhibitor (PhosSTOP™, Roche Diagnostics GmbH, Mannheim, Germany)) was applied, and fractions were collected (2 × 1 mL). Next, 1 mL of elution buffer 2 (50 mM citric acid, pH 2.4 adjusted with 1 N NaOH, 100 mM NaCl, 1 mM MgCl_2_, 300 mM D-mannose) was applied, and the fraction was collected (1 mL). SDS-PAGE and protein gel blot analysis of the purified fractions were performed as stated above. HvHorcH was mainly found in the first fraction of elution 1, which was stored at −20 °C until further usage.

### 4.11. Generation of Transgenic Arabidopsis Plants

For the generation of transgenic *A. thaliana* plants, the Col-0 ecotype was used as the genetic background. Plants were grown in a greenhouse under long-day (16 h of light/8 h of darkness) conditions at 22 °C. For the generation of pGSTu:HvHorcH;eGFP transgenic plants, 1.36 kb pGSTu25 promoter (pGSTu25) and eGFP were subcloned together with HvHorcH ORF (see Section 4.6) into Gateway^®^ Donor vectors. All three components were introduced into pB7m34GW expression vector using LR Clonase™ II Plus (Thermo Scientific, Waltham, MA, United States). Constructs were subsequently introduced into *Agrobacterium tumefaciens* strain GV3101, and wild-type plants were transformed using the floral dip method [66]. Transgenic lines were selected on medium containing phosphinothricin (PPT).

## Figures and Tables

**Figure 1 ijms-22-10248-f001:**
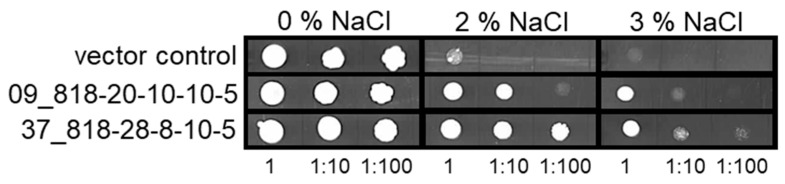
Salt stress tolerance screening of transfected YSH818 *Δhog1* yeast cells plated on agar containing increasing salt concentrations. Yeast cells were either transfected with cDNA expression vector alone (upper row) or vectors containing two *HvHorcH* (HORVU7Hr1G059330) cDNAs. The respective inocula were deposited in tenfold dilution steps, from left to right for each strain and salt condition.

**Figure 2 ijms-22-10248-f002:**
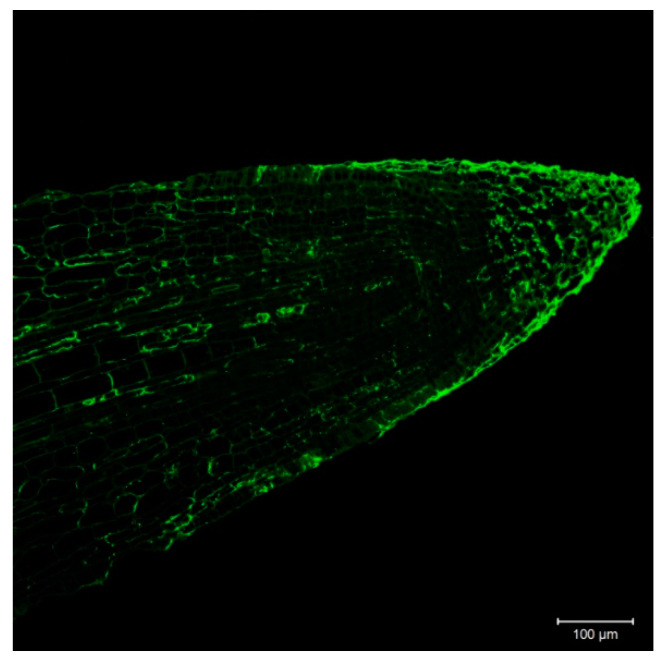
Immunolocalization of HvHorcH on barley cv. Morex root tips after salt stress treatment. HvHorcH was found mainly in the root cap.

**Figure 3 ijms-22-10248-f003:**
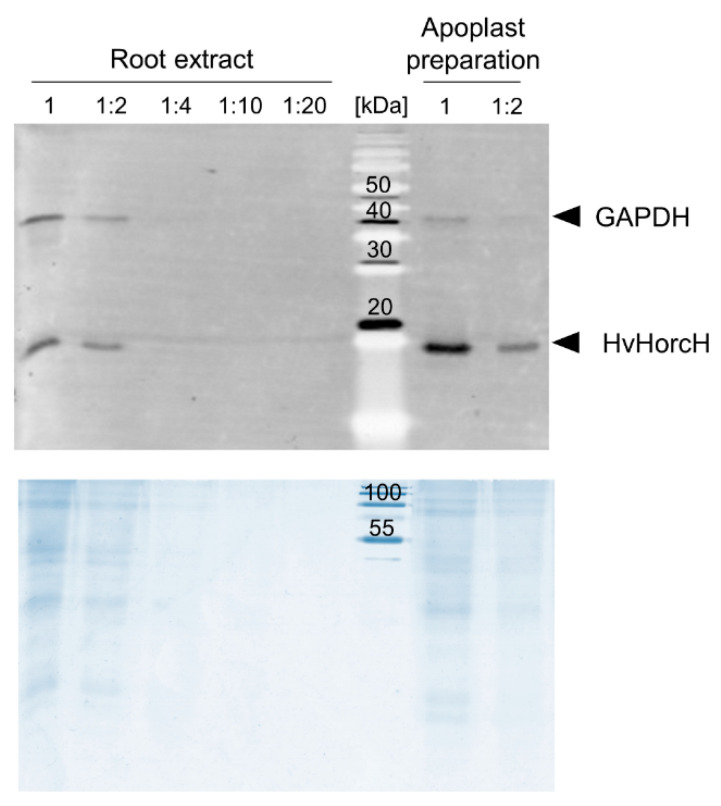
Detection of HvHorcH in the root apoplast. Anti-GAPDH and anti-HvHorcH stained Western blot (top) of an SDS-PAGE (bottom: Coomassie stained) with dilution series of cv. Morex native root extracts or apoplastic fluids. For the apoplastic fluid preparation, an infiltration vacuum of 20 mPa was applied with a vacuum pump under manometer control. All extracts were prepared from salt stress (150 mM NaCl)-treated roots. Two µg of protein of root extract or apoplast preparation was loaded to the SDS-PAGE as undiluted sample and then diluted as indicated in the figure.

**Figure 4 ijms-22-10248-f004:**
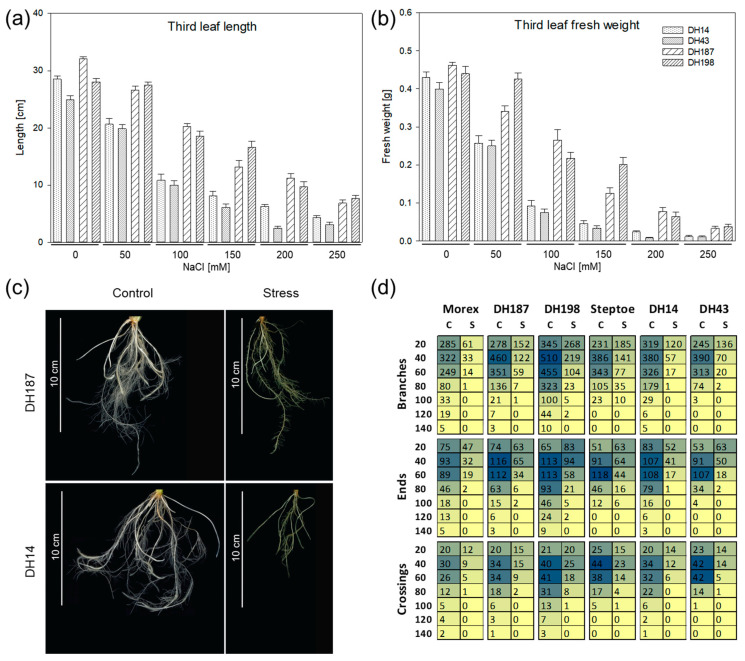
The effect of NaCl on the seedling growth and root growth of salt-tolerant (cv. Morex, DH 187, DH198) and salt-sensitive (cv. Steptoe, DH14, DH43) accessions of the Steptoe × Morex mapping population. (**a**,**b**) Salt tolerance was recorded as indicated by biomass production of the third leaf. The data represent the means of 17–20 plants per treatment, with the standard error shown as error bars. Bars filled with dots indicate salt-sensitive accessions, while bars filled with stripes represent tolerant ones. (**c**) Representative photographical images of control plants and plants under salt stress from DH187 and DH14. (**d**) Specific features of root architecture including branches, ends, and crossing points were measured in distinct circular segments centered on the root base. Values refer to the distance from the root base in mm (from 20 to 140). Mean values are shown in a color code representation. Blue indicates high numbers, while beige represents low values.

**Figure 5 ijms-22-10248-f005:**
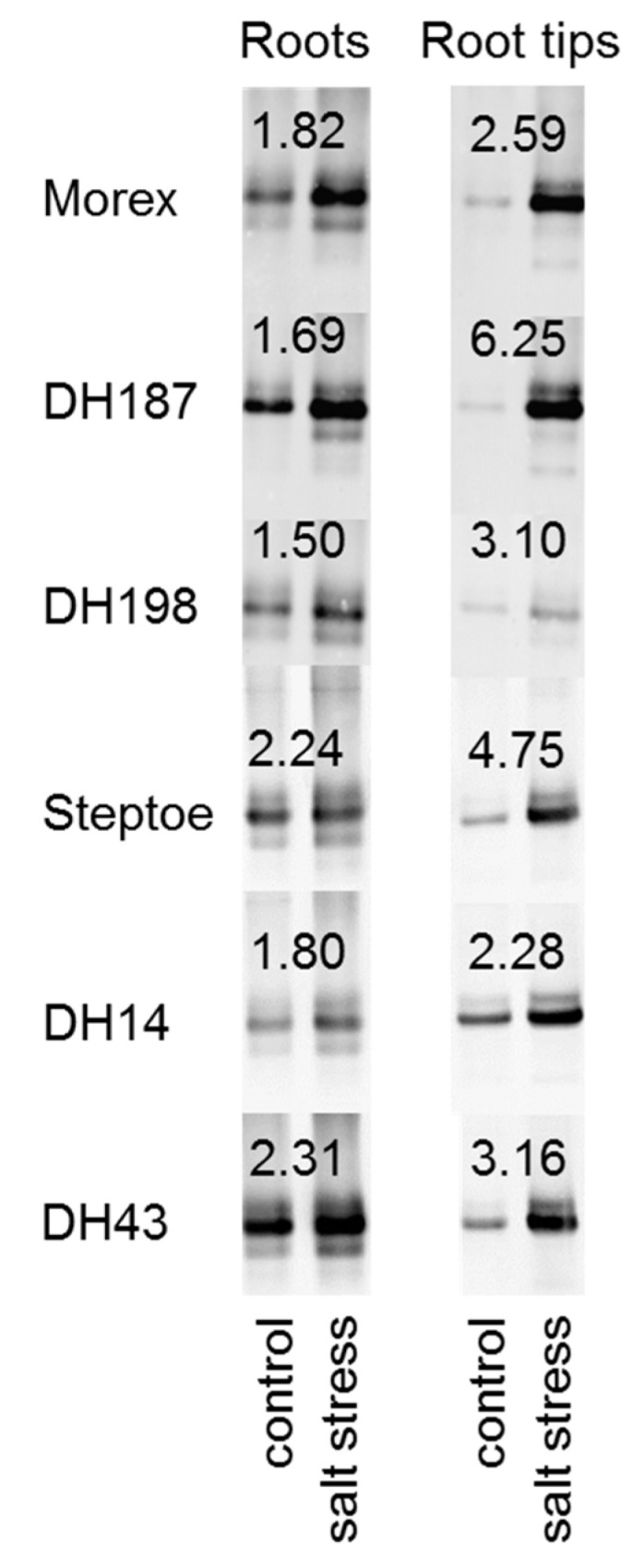
HvHorcH abundance in roots and root tips of tolerant (cv. Morex, DH 187, DH198) or sensitive (cv. Steptoe, DH14, DH43) barley accessions with contrasting salt stress response. Equal amounts of total protein (2 µg) were separated by SDS-PAGE on 15% gels, and HvHorcH was immunologically detected by protein gel blot analysis. Pixel intensities of detected bands were integrated. The mean fold change between control and salt stress (150 mM NaCl) conditions calculated from three independent experiments is given on top of the bands. Representative images from one experiment are shown.

**Figure 6 ijms-22-10248-f006:**
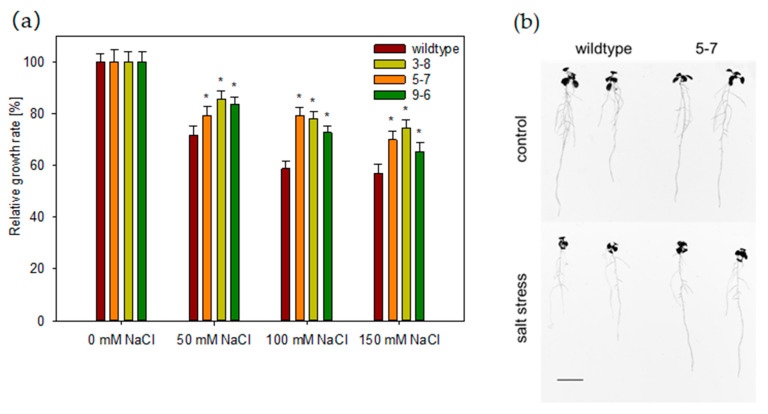
Salt stress response of *Arabidopsis thaliana* plants expressing *HvHorcH* in root tips. (**a**) Quantification of the relative rosette growth rate in *A. thaliana* wildtype Col-0 and *HvHorcH* transgenic plants (lines 3-8, 5-7, 9-6) under three different levels of salt stress, grown in soil culture. The data represent the means of about 80 plants per treatment and genotype, originating from three independent experiments, with the standard error shown as error bars. Asterisks (*) indicate significant differences between the wildtype Col-0 and *HvHorcH* transgenic plants (*t*-test, *p* < 0.01) (**b**) Root phenotypes of a representative transgenic line pGSTu25:HvHorcH:GFP #5-7 compared with the wildtype after 7 d of growth on an agar plate in the absence (top) or presence (bottom) of 50 mM NaCl (bar = 1 cm).

**Table 1 ijms-22-10248-t001:** Barley cDNAs conferring salt tolerance to a salt-sensitive yeast mutant. Given are the BLAST result (barlex CDS_HC May2016 database using blastx), the putative annotation and putative function of the gene, and the number of isolates from yeast transformants.

Annotation	Gene	Score (Bits)	E Value	Function	Number of Independent Yeast Transformants
jacalin-related lectin 31 (HvHorcH)	HORVU7Hr1G059330	296	7 × 10^−103^	carbohydrate binding	10
cellulose-synthase like D2	HORVU2Hr1G042350	502	2 × 10^−174^	cellulose biosynthesis	6
peptide-N(4)-(N-acetyl-β-glucosaminyl)asparagine amidase	HORVU2Hr1G048680	507	3 × 10^−177^	protein deglycosylation	3
WD-40 repeat family protein	HORVU3Hr1G115170	376	4 × 10^−128^	translation	3
phytanoyl-CoA dioxygenase domain-containing protein 1	HORVU4Hr1G007050	439	3 × 10^−156^	fatty acid metabolism	3
unknown function	HORVU4Hr1G070280	64	8 × 10^−14^	unknown	3
histone H2A 6	HORVU7Hr1G100110	171	1 × 10^−52^	DNA binding	3
30S ribosomal protein S11	HORVU7Hr1G104220	256	8 × 10^−88^	translation	3
exocyst complex component 6	HORVU0Hr1G006630	367	2 × 10^−122^	vesicle transport	2
membrane steroid binding protein 1	HORVU1Hr1G045630	276	8 × 10^−93^	vesicle transport	2
histone H2A 2	HORVU2Hr1G043860	145	1 × 10^−44^	DNA binding	2
unknown function	HORVU7Hr1G032340	159	2 × 10^−48^	unknown	2
cytochrome P450 superfamily protein	HORVU1Hr1G069310	124	2 × 10^−43^	gibberellin catabolic process	1
GDSL esterase/lipase	HORVU2Hr1G025800	280	4 × 10^−93^	fatty acid metabolism	1
adenine nucleotide alpha hydrolases-like superfamily protein	HORVU2Hr1G042480	69	4 × 10^−16^	stress response	1
ribosomal protein S24e family protein	HORVU4Hr1G058010	207	1 × 10^−67^	unknown	1
potassium transporter 26	HORVU4Hr1G058080	39	7 × 10^−04^	ion transport	1
26S proteasome non-ATPase regulatory subunit 4 homolog	HORVU4Hr1G063820	370	3 × 10^−126^	proteolysis	1
ATP-dependent zinc metalloprotease FtsH 2	HORVU5Hr1G063340	68	2 × 10^−15^	proteolysis	1
peptidyl-prolyl cis-trans isomerase	HORVU6Hr1G012570	295	3 × 10^−102^	protein folding	1
Rad23 UV excision repair protein family	HORVU7Hr1G042100	221	4 × 10^−70^	mRNA catabolism	1

## Data Availability

The data presented in this study are available in the article and Appendix A.

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
