# Peer review of "The Jacalin-Related Lectin HvHorcH Is Involved in the Physiological Response of Barley Roots to Salt Stress"

_ijms, 2021, doi:10.3390/ijms221910248_

Round 1

Reviewer 1 Report

This article is significant that the expression of HvHorcH is involved in the adaptation of plants to salinity.

I checked your manuscript.

My comment is below.

  1. I can't see the rightmost column in Table1.
  2. It had better that SDS-PAGE results need a little more detail in Figure 3.

Author Response

Kind regards,

Katja Witzel, on behalf of all authors

Reviewer 2 Report

As a general note, it seems that the manuscript was submitted as incomplete or at least not the final version. I see many yellow, green highlights that were created by one of the coauthors are left behind. Table 1 has at least one column that is not fully visible. At many places throughout the text, I see two full stops giving me the impression that it might not be the final version of the manuscript.

Since the cDNA library that was used in the investigation represented seedling stage, I would suggest mentioning it at appropriate places such as in the title, abstract.

Authors used non-transgenic Col-0 as control. Ideally, in such experiments empty vector is recommended for control (similar approach that the authors used in their Yeast experiment) to rule out any effect(s) introduced by transformation process. Authors may like to address this concern.

Author Response

(The authors gave the same response as above.)

Round 2

Reviewer 2 Report

Authors have addressed the comments. The revised version looks good to proceed for its publication.